# Developing and validating a cross-cultural competence scale for Japanese nurses

**Masako Sakamoto** [1] *, **Shigemi Iriyama**[1], **Yu Mon Saw**[2]

**1** Department of Integrated Health Sciences, Nagoya University Graduate School of Medicine, Nagoya, Japan, **2** Department of Community and Global Health, University of Tokyo, Tokyo, Japan

* sakamoto.masako.w3@s.mail.nagoya-u.ac.jp

## Abstract

In recent years, Japan has experienced a significant increase in the number of foreign students and workers entering the country. This has resulted in a vast number of international patients in medical facilities. This shift emphasizes the immediate need for Japanese nurses who are both clinically proficient and culturally attuned. In response, our research developed and validated the Cross-cultural Competence Scale for Japanese Nurses (CCCSJN) to better equip nurses for diverse patient care. We conducted a cross-sectional study in Japan's general hospitals using anonymous questionnaires with nurses and midwives. The scale, developed from data from 394 nurses, underwent both qualitative and quantitative evaluations to define its construct. We analyzed the data using exploratory factor analysis, criterion-related validity, internal consistency, and test-retest reliability, confirming the scale's reliability and validity. The exploratory analysis revealed five factors: "cross-cultural understanding," "cross-cultural communication ability," "motivation for cross-cultural nursing," "cooperation with multiple professions," and "respect for foreign patients." These factors explained 50.92% of the total variance. Cronbach's α for the CCCSJN was 0.94, and the test-retest reliability correlation was 0.77. The construct validity, criterion-related validity, internal consistency, and test-retest reliability of the CCCSJN were verified. The CCCSJN can be used to assess the cross-cultural competencies of Japanese nurses and identify what skills need to be mastered, leading to improved cross-cultural competence and care.

## Introduction

There has been a noticeable upward trend in the influx of foreign workers and students into Japan in recent years, resulting in an increase in the number of foreign patients seeking care in medical facilities. According to a Ministry of Health, Labour and Welfare report, the number of foreign patients per medical facility more than doubled between 2018 and 2020, from 24.3% to 50.2% [1]. In 2021, the number of medical facilities receiving foreign patients reached a record high, with 50.1% of all facilities responsible for accommodating this influx [1].

Despite the impact of the COVID-19 pandemic, the number of foreign residents has not decreased significantly, remaining at approximately 2.82 million in 2021 [2] or 2.5% of the overall population. Since April 29, 2023, restrictions on foreign entry into Japan have eased, international travel has increased, and the number of foreign patients visiting Japanese medical

**Data Availability Statement:** The dataset, which is the minimal basis for the study, was uploaded as a public repositor by the Nagoya University Repository. The uploaded data set is as follows.
Table 1    https://doi.org/10.18999/2008709

Table 2 https://doi.org/10.18999/0002008753
Table 3 https://doi.org/10.18999/2008754 Table 4
https://doi.org/10.18999/2008755 Table 5 https://
doi.org/10.18999/2008756 Table 6 https://doi.org/
10.18999/2008757.

**Funding:** The author(s) received no specific
funding for this work.

**Competing interests:** The authors have declared
that no competing interests exist.

facilities is expected to increase. Medical facilities that previously had minimal or no contact with foreign patients must promptly establish a system to accommodate them [3]. Similarly, there is an increasing demand for Japanese nurses who can effectively care for foreign patients.

Culturally sensitive care must be provided to effectively address the demands of foreign patients from various cultural backgrounds [4, 5]. Although 72.5% of Japanese nurses have experience in caring for foreign patients [6], these nurses acknowledged that they have not been able to provide adequate care. Japanese nurses often experience anxiety and difficulty in language and communication [7–10], as well as difficulties in addressing cultural aspects such as lifestyle, diet, and religious practices [11, 12].

To ensure the safe and uncomplicated delivery of medical care to foreign patients, nurses' competencies in interacting with culturally diverse individuals is critical. This goes beyond fundamental communicative skills, necessitating a sophisticated comprehension of various cultural factors such as lifestyle choices, dietary preferences, and religious practices [13]. Unfortunately, Japanese nurses are often hesitant to provide care to foreign patients because of the bewilderment caused by differences in communication, culture, and customs. A lack of cross-cultural competence can affect the quality of nursing care [14] and reduce patient satisfaction and safety [15]. However, if nurses can properly assess their own cross-cultural competence, they can visualize what competencies are lacking. This increased self-awareness can improve the quality of care they provide to foreign patients.

Globally, numerous studies on nursing care for foreign patients have produced various scales to measure nurses' cross-cultural competence. Widely used scale concepts include three-dimensional models, such as the scale of beliefs and attitudes, knowledge, and skills developed by Suh et al. in the United States [13]. While several measures of cross-cultural competence have been properly tested and shown to be reliable, many lack the necessary validation [16, 17]. It is essential to employ appropriately validated scales to assess nurses' cross-cultural competencies.

Currently, only a few scales assess the cross-cultural competence of Japanese nurses. These include Cultural Competence in Nursing (CCN) [18] and the Japanese version of the Caffrey Cultural Competence Health Services (J-CCCHS) [6]. The CCN was first developed in Japan by Sugiura in 2003, referring to the Campinha–Bacote conceptual model [19, 20]. Although construct validity was shown, the Cronbach's α for one factor was low (0.64). Additionally, the number of questions was large (n = 46), raising concerns about the burden on participants. Therefore, the validity and reliability of the scale needs further examination. Further, the J-CCCHS was developed through a large-scale survey conducted by Noji in 2017. Structural equation modeling was used for the analysis, but the scale is not yet practical owing to insufficient model fit [6]. Further improvements are urgently needed to create a valid and reliable measure for Japanese nurses.

The ultimate purpose of developing and validating the Cross-cultural Competence Scale for Japanese Nurses (CCCSJN) was to create a tool that accurately visualizes the current state of cross-cultural competency among Japanese nurses. This will provide valuable data to guide improvements in cross-cultural competency, enhancing the ability of healthcare practitioners to effectively serve Japan's growing and increasingly diverse foreign population, which has been expanding. As Japan continues to welcome more foreigners, cultural competence in healthcare is becoming increasingly essential, and this study helps to close that gap.

## Materials and methods

### Preliminary study

To develop the CCCSJN, it was necessary to use questionnaire items that reflect the characteristics of Japanese nurses in caring for patients with different cultural backgrounds. Therefore,

a preliminary qualitative study was conducted. The results were then used as items in the questionnaire.

We conducted 30-minute semi-structured interviews based on interviews with 14 Japanese nurses with experience in caring for foreign patients after obtaining informed consent. We analyzed the concept of cross-cultural responsiveness from data recorded by IC recorders and extracted eight categories: consideration for foreign patients, motivation to care for foreign patients, ingenuity in explanation, utilization of social resources, confirmation, dignity, financial support, and collaboration. We then drafted the 40-item CCCSJN. The CCSJN used a five-point Likert scale ranging from "not at all" to "very true".

To assess content validity, four university faculty members working as cross-cultural nursing researchers examined the clarity of each item. The three items related to utilization of social resources, confirmation, and dignity were deleted because they were redundant with other questions or were difficult to understand, leaving 37 items in the final draft. The overall content validity index of the scale was 0.96.

## Study design and study duration

We conducted a cross-sectional study using an anonymous self-administered questionnaire between August 2022 and January 2023.

## Study participants

We examined nurses and midwives in general hospitals located in four cities in Aichi Prefecture with many foreign residents. The selection criteria for participants were nurses and midwives of Japanese nationality. Head nurses of outpatient departments and wards were excluded because administrative responsibilities greatly expanded their roles beyond simply providing care.

## Sampling procedures and data collection

Japanese nurses were recruited through convenience sampling from six medical facilities with more than 200 beds. We selected Aichi Prefecture as the site for this study because it has the second-largest number of foreign residents after Tokyo. Consequently, nurses working in Aichi Prefecture were speculated to have more experience caring for foreign patients than those in other prefectures. In total, there were 165 potential medical facilities. However, the number of eligible medical facilities was 27, excluding psychiatric, oncology, pediatric, and recuperation facilities and medical facilities with less than 200 beds. Research cooperation was requested in writing from the nursing directors of these 27 medical facilities, and consent was obtained from the nursing directors of six medical facilities. Nursing directors in the outpatient and ward departments distributed study request forms, two anonymous self-report questionnaires, and post-paid envelopes to the participants. We collected data by mail.

## Definition of "cross-cultural competence"

"Cross-cultural competence" is the ability to provide appropriate care to foreign patients with different cultural backgrounds. It was defined with reference to the three-dimensional model of Suh et al. [13]. This ability requires cross-cultural understanding, communication skills, and respect for foreign patients.

## Study measures

**Social critical thinking orientation (SCTO).** To verify the criterion-related validity of the CCCSJN, we used two SCTO subscales "understanding diversity" and "understanding others" developed by Nakanishi et al. [21]. The SCTO comprises 27 items and seven factors: emphasis on logic and evidence, discrediting, understanding others, understanding diversity, authenticity, understanding key points, and determination. Permission to use the SCTO was obtained from the authors via email.

The subscales "understanding diversity" and "understanding others" correspond to the "power to understand people" and the "power to practice people-centered care" among the elements of nursing practical ability [22]. Therefore, the two subscales were adopted to infer that they were related to the cross-cultural ability of Japanese nurses to provide the necessary care to foreign patients. Both "understanding diversity" and "understanding others" comprise four items and are rated on a seven-point Likert scale from "not at all applicable" (1) to "very applicable" (7). We calculated the sum score, with higher scores indicating a higher degree of understanding diversity and others.

**Japanese intercultural sensitivity scale (J-ISS).** To verify the criterion-related validity of the CCCSJN, we used a subscale of the J-ISS [23]: "positive feelings for different cultures." The ISS was developed by Chen and Starosta [24], and the J-ISS was developed by Suzuki et al. [20]. The J-ISS comprises 22 items and three concepts: "positive feelings toward different cultures," "ambivalent feelings toward different cultures," and "negative feelings toward different cultures." Permission to use the J-ISS was obtained from the authors via email.

"Positive feelings toward different cultures" comprises items with positive emotional content, such as actively wanting to know about cultural differences or enjoying cultural differences [23]. We speculated that this concept is related to cross-cultural competence needed by Japanese nurses caring for foreign patients and was adopted as the criterion-related validity of the CCCSJN. The J-ISS subscale "positive feelings toward different cultures" comprises 10 items with a five-point Likert scale ranging from "not at all" (1) to "very true" (5). We calculated the sum of the scores, with high scores indicating a high degree of positivity.

**Socio-demographic characteristics.** We collected socio-demographic information such as age, years of work experience, department affiliation, nursing qualifications, and educational background.

**Cross-cultural experience in nursing.** We collected data on the frequency of foreign patient care, experience of living abroad for more than a month, contact with foreigners, foreign language proficiency, and experience in cross-cultural education.

## Statistical analysis

Age and years of nursing experience were not normally distributed and thus were calculated with median and interquartile range (IQR).

The construct validity of the CCCSJN was assessed using exploratory factor analysis. An item analysis was performed beforehand to examine ceiling and floor effects and item-total correlations. The principal factor method was selected for factor extraction and promax rotation was used because there was a correlation between the subscales. The statistical criteria determined the extraction of factors with eigenvalues of $\geq 1$. Based on these analyses, items with loadings $> 0.40$ were considered significant and factors were extracted.

To verify the criterion-related validity of the CCCSJN, the relationships between the subscales of the CCCSJN and the SCTO ("understanding others" and "understanding diversity") and between the subscales of the CCCSJN and the J-ISS ("positive feelings toward different cultures") were examined using Spearman rank correlation coefficients.

The reliability (internal consistency) of the CCCSJN was determined by Cronbach's alpha coefficients. The test-retest reliability of the first and second surveys was indicated by intraclass correlation coefficients (ICCs).

Data were analyzed using SPSS version 22.0 (IBM SPSS Inc.).

### Ethical considerations

We obtained ethical approval from the ethics committee of Nagoya University School of Medicine (no. 22–108) and the six medical facilities. The study purpose was fully explained to all participants, and participants were not obliged to participate in this study and could withdraw at any time. A consent box was provided at the beginning of the questionnaire, and participants were asked to check the box if they agreed to participate.

## Results

### Descriptive statistics

In the main study, 406 out of 504 (72.2% effective response rate) responded by mail, and 394 out of 406 (97.0% effective response rate) responded to the CCCSJN questionnaire. All participants were women. The median age of the nurses was 33.0 years (IQR: 26–44), and their nursing experience was 9.8 years (IQR: 4.6–19.7). Of these, 330 (83.8%) completed their academic career at vocational schools and 51 (13.0%) had graduated from universities or graduate schools. There were 317 (80.5%) nurses working at city hospitals and 19 (4.8%) working at university hospitals (Table 1).

There were 12 (3.0%) nurses who had at least one month's experience with living overseas, while five (1.3%) had overseas training or work experience. Fifty-four (13.7%) nurses had contact with foreigners other than patients in their daily lives, and 22 (5.6%) nurses had foreign skills equivalent to daily conversations. Ninety-five (24.2%) nurses had cross-cultural understanding and learning experiences, while 325 nurses (82.5%) had cared for one or more foreign

**Table 1. Japanese nurses' socio-demographics (N = 394).**

|  |  | n | % | Median | IQR |
|---|---|---|---|---|---|
| **Age (years)** |  |  |  | 33 | 26–44 |
| **Length of clinical experience (years)** |  |  |  | 9.8 | 4.6–19.7 |
| **Qualification** | Nurse | 372 | 94.4 |  |  |
|  | Midwife | 22 | 5.6 |  |  |
| **Educational background** | Diploma | 330 | 83.7 |  |  |
|  | Bachelor's degree | 44 | 11.2 |  |  |
|  | Advanced diploma | 9 | 2.3 |  |  |
|  | Master's degree | 7 | 1.8 |  |  |
|  | Associate degree | 4 | 1.0 |  |  |
| **Department** | Inpatient | 321 | 81.5 |  |  |
|  | Outpatient | 73 | 18.5 |  |  |
| **Number of nurses by employment facility** | Municipal hospital | 317 | 80.5 |  |  |
|  | Consumers' co-operative hospital | 46 | 11.7 |  |  |
|  | University hospital | 19 | 4.8 |  |  |
|  | Public interest incorporated foundation hospital | 12 | 3.0 |  |  |

IQR, interquartile range.

https://nagoya.repo.nii.ac.jp/records/2008709

**Table 2. Japanese nurses' cross-cultural experiences (N = 394).**

|  |  | n | % | Median | IQR |
|---|---|---|---|---|---|
| **Stay abroad[a] (≥ 1 month)** | Yes | 12 | 3.1 |  |  |
|  | No | 380 | 96.9 |  |  |
| **Abroad nursing experience (training/work)** | Yes | 5 | 1.3 |  |  |
|  | No | 389 | 98.7 |  |  |
| **Contact with foreigners other than foreign patients** | Yes | 54 | 13.7 |  |  |
|  | No | 340 | 86.3 |  |  |
| **Conversational proficiency in a foreign language** | Yes | 22 | 5.6 |  |  |
|  | No | 372 | 94.4 |  |  |
| **Cross-cultural learning[b]** | Yes | 95 | 24.2 |  |  |
|  | No | 298 | 75.8 |  |  |
| **Experience of foreign patient care within the year** | Yes | 325 | 82.5 |  |  |
|  | No | 69 | 17.5 |  |  |
| **Number of foreign patient care experiences within the year[c]** |  |  |  | 5 | 3–10 |

IQR, interquartile range. [a]Missing value: 2. [b]Missing value: 1. [c]Missing value: 76.

https://nagoya.repo.nii.ac.jp/records/2008753

patients in the past year. The median number of times nurses provided care for foreign patients in the past year was five (IQR: 3–10; Table 2).

## Validity and reliability of the CCCSJN

**CCCSJN construct validity.** To confirm the construct validity of the CCCSJN, item analysis was performed before the exploratory factor analysis. The distribution showed a ceiling effect for Q19 (5.09), Q20 (5.01) and Q21 (5.08), while Q4 showed a floor effect (0.75). The I-T correlation was less than 0.30 for Q4 (0.06), Q9 (0.23), Q20 (0.29), and Q21 (0.21). Therefore, five of the 37 items were excluded from the CCCSJN (Table 3).

In the first exploratory factor analysis, Q5, Q18, Q22, and Q24 were excluded because their factor loadings were less than 0.4. The factor loading for Q8 was 0.38, but it was borderline and similar to the factor concepts in questions 6 and 7; therefore, it was not excluded and was reserved. In the second exploratory factor analysis, all questionnaire items were above 0.40. The factor loading of Q17 was 0.65, but it was excluded because it was one factor and one item (Table 4). To ensure that exclusion criterion of less than 0.40 was reasonable, an exploratory factor analysis was performed by excluding items with a factor loading of less than 0.35. The result was a six-factor structure with 29 items. However, one factor included only two items: Q17 and Q18. The Cronbach's α was low (0.53).

A third exploratory factor analysis was performed, and the criterion was less than 0.40. Finally, a five-factor structure comprising 27 items was adopted. The model fit showed that the validity of the Kaiser–Meyer–Olkin sample was 0.918, and Bartlett's sphericity test showed a good fit (p < .001). The cumulative contribution rate was 50.92 (Table 5).

The first to fifth factors contributed 32.53%, 6.54%, 5.33%, 3.69%, and 2.84% of the variance on cultural competence, respectively. The factor loadings were 0.62–0.84 for eight items on the first factor, 0.40–0.71 for five items on the second factor, 0.44–0.71 for six items on the third factor, 0.55–0.79 for five items on the fourth factor, and 0.42–0.92 for three items on the fifth factor. We named the first factor "cross-cultural understanding," the second factor "cross-cultural communication ability" the third factor "motivation for cross-cultural nursing," the fourth factor "cooperation with multiple professions," and the fifth factor "respect for foreign patients".

**Table 3. Item analysis of the CCCSJN for Japanese nurses (N = 394).**

| | Question items | Mean | SD | CE | FE | I-T |
|---|---|---|---|---|---|---|
| Q1 | I treat foreign patients with respect. | 4.02 | 0.72 | 4.74 | 3.30 | 0.56 |
| Q2 | I frequently talk to foreign patients. | 3.40 | 0.90 | 4.30 | 2.50 | 0.60 |
| Q3 | I greet foreign patients in their native languages as much as possible. | 2.68 | 1.11 | 3.79 | 1.57 | 0.38 |
| Q4 | When foreign patients are admitted to a ward, I always introduce them to the patients in the same room. | 1.69 | 0.94 | 2.63 | 0.75 | 0.06 |
| Q5 | I try not to show foreign patients that I am in a hurry. | 3.15 | 0.86 | 4.01 | 2.30 | 0.34 |
| Q6 | When I cannot satisfy the wishes of a foreign patient, I explain the reason and obtain his/her understanding. | 3.54 | 0.84 | 4.38 | 2.70 | 0.53 |
| Q7 | When a foreign patient asks me a question about nursing care, I provide a detailed explanation. | 3.61 | 0.80 | 4.42 | 2.81 | 0.58 |
| Q8 | I listen to foreign patients carefully and try to understand their thoughts. | 3.86 | 0.70 | 4.57 | 3.16 | 0.66 |
| Q9 | When I know I'll be dealing with foreign patients in advance, I try to research the culture of the country they come from. | 2.22 | 1.03 | 3.25 | 1.19 | 0.23 |
| Q10 | I try to provide the same care to foreign patients as I do to Japanese patients. | 4.11 | 0.77 | 4.88 | 3.34 | 0.51 |
| Q11 | I apply the experiences I have learned from foreign patients in my nursing care. | 3.16 | 0.86 | 4.02 | 2.31 | 0.52 |
| Q12 | When there is a difference between the wishes of foreign patients and the rules of the hospital, I think of other ways to address the issue together with the patient. | 2.87 | 0.92 | 3.78 | 1.95 | 0.47 |
| Q13 | I want to be actively involved in the care of foreign patients. | 2.87 | 0.96 | 3.83 | 1.91 | 0.49 |
| Q14 | I take time to explain matters to foreign patients. | 3.45 | 0.85 | 4.30 | 2.60 | 0.63 |
| Q15 | I speak slowly to foreign patients in simple Japanese. | 4.13 | 0.65 | 4.77 | 3.49 | 0.45 |
| Q16 | I use gestures to explain things to foreign patients who do not understand Japanese. | 4.27 | 0.67 | 4.94 | 3.60 | 0.52 |
| Q17 | I explain to foreign patients by showing them illustrations and drawings. | 3.44 | 1.01 | 4.45 | 2.43 | 0.35 |
| Q18 | I use a comparison chart for foreign patients in their native language or in an understandable foreign language and Japanese. | 3.49 | 1.12 | 4.61 | 2.37 | 0.36 |
| Q19 | I use translation devices and translation apps for foreign patients. | 4.10 | 0.99 | 5.09 | 3.11 | 0.34 |
| Q20 | I get the cooperation of family and friends who understand Japanese. | 3.99 | 1.02 | 5.01 | 2.97 | 0.29 |
| Q21 | I always use medical interpreters when I give IC or important explanations to foreign patients. | 3.99 | 1.09 | 5.08 | 2.90 | 0.21 |
| Q22 | I check the thoughts and intentions of foreign patients before providing care. | 3.45 | 0.87 | 4.32 | 2.58 | 0.58 |
| Q23 | I observe the facial expressions and reactions of foreign patients to see if they understand what I want to convey. | 4.06 | 0.63 | 4.68 | 3.44 | 0.66 |
| Q24 | I ask questions to make sure foreign patients understand what I have tried to convey. | 3.72 | 0.81 | 4.53 | 2.91 | 0.39 |
| Q25 | I understand that the cultures of foreign patients may make them have different views regarding health and disease matters. | 3.52 | 0.88 | 4.40 | 2.64 | 0.62 |
| Q26 | I understand that different cultures have different values. | 3.89 | 0.77 | 4.66 | 3.12 | 0.69 |
| Q27 | I understand that foreign patients' religion or beliefs may dictate traditional diets or dietary restrictions. | 4.03 | 0.73 | 4.76 | 3.30 | 0.69 |
| Q28 | I understand that foreign patients may have behavioral restrictions or rules according to their religion. | 3.95 | 0.74 | 4.69 | 3.21 | 0.69 |
| Q29 | I understand that culturally sensitive nursing care must be provided to all foreign patients | 3.92 | 0.77 | 4.69 | 3.15 | 0.70 |
| Q30 | I understand that foreign patients who are here for a short period of stay may feel anxious about their hospital stay. | 3.88 | 0.82 | 4.71 | 3.06 | 0.68 |
| Q31 | I understand that the health care system in foreign patients' country of origin is different. | 3.50 | 0.94 | 4.44 | 2.56 | 0.61 |
| Q32 | I understand that foreign patients may have financial concerns. | 3.79 | 0.78 | 4.57 | 3.01 | 0.63 |
| Q33 | I share information with the staff in my office in order to understand foreign patients. | 3.60 | 0.84 | 4.44 | 2.76 | 0.63 |
| Q34 | I have detailed information on the daily lives of foreign patients in my medical records. | 3.15 | 0.85 | 4.00 | 2.30 | 0.52 |
| Q35 | I do not handle foreign patients by myself, I collaborate with my colleagues in the office. | 3.79 | 0.75 | 4.54 | 3.04 | 0.62 |
| Q36 | I cooperate with staff from various professions to meet the wishes of foreign patients. | 3.75 | 0.75 | 4.51 | 3.00 | 0.59 |
| Q37 | With the consent of the foreign patient, I share information with supporters. | 3.55 | 0.81 | 4.36 | 2.74 | 0.54 |

CCCSJN, Cross-cultural Competence Scale for Japanese Nurses; SD, standard deviation; CE, ceiling effect; FE, floor effect; I-T, i-t correlation.

https://nagoya.repo.nii.ac.jp/records/2008754

The results were obtained by an exploratory factor analysis following the analysis of 37 items in the draft CCCSJN. To confirm the factor structure, an exploratory factor analysis was performed on the draft before item analysis. The following 10 items with factor loadings < 0.4 were excluded: Q5, Q14, and Q17–Q24. The four items related to the utilization of social resources were included. A five-factor structure comprising 27 items was clarified.

**Table 4. Factor loadings on the CCCSJN for Japanese nurses after items analysis (N = 394).**

| | Items | 1 | 2 | 3 | 4 | 5 | 6 |
|---|---|---|---|---|---|---|---|
| Q26 | I understand that different cultures have different values | **0.86** | 0.11 | 0.03 | -0.11 | -0.12 | 0.05 |
| Q28 | I understand that foreign patients may have behavioral restrictions or rules according to their religion. | **0.86** | -0.07 | 0.19 | -0.11 | -0.03 | 0.00 |
| Q27 | I understand that foreign patients' religion or beliefs may dictate traditional diets or dietary restrictions | **0.82** | -0.07 | 0.22 | -0.11 | -0.04 | -0.01 |
| Q31 | I have detailed information on the daily lives of foreign patients in my medical records. | **0.76** | 0.06 | -0.27 | 0.14 | 0.00 | -0.10 |
| Q29 | I understand that culturally sensitive nursing care must be provided to all foreign patients. | **0.75** | -0.08 | 0.12 | 0.03 | 0.06 | -0.04 |
| Q32 | I understand that foreign patients may have financial concerns. | **0.69** | -0.07 | -0.13 | 0.19 | 0.11 | -0.14 |
| Q25 | I understand that the cultures of foreign patients may make them have different views regarding health and disease matters. | **0.65** | 0.28 | -0.20 | 0.00 | -0.11 | 0.13 |
| Q30 | I understand that foreign patients who are here for a short period of stay may feel anxious about their hospital stay. | **0.64** | -0.13 | 0.04 | 0.09 | 0.13 | 0.09 |
| Q13 | I want to be actively involved in the care of foreign patients. | -0.04 | **0.85** | 0.04 | -0.04 | -0.21 | 0.03 |
| Q2 | I frequently talk to foreign patients. | -0.15 | **0.62** | 0.25 | 0.04 | 0.03 | -0.15 |
| Q3 | I greet foreign patients in their native languages as much as possible. | -0.03 | **0.53** | -0.07 | -0.10 | 0.03 | 0.14 |
| Q11 | I apply the experiences I have learned from foreign patients in my nursing care. | 0.15 | **0.49** | -0.09 | -0.01 | -0.03 | 0.10 |
| Q12 | When there is a difference between the wishes of foreign patients and the rules of the hospital, I think of other ways to come to terms with them together with foreign patients. | 0.07 | **0.44** | -0.20 | 0.06 | 0.15 | 0.14 |
| Q14 | I take time to explain to foreign patients. | 0.11 | **0.43** | 0.17 | 0.00 | 0.10 | 0.01 |
| Q22 | I check the thoughts and intentions of foreign patients before providing care. | 0.07 | 0.07 | 0.07 | 0.07 | 0.07 | 0.07 |
| Q24 | I ask questions to make sure foreign patients understand what I have tried to convey. | 0.26 | 0.28 | 0.13 | 0.07 | 0.11 | 0.00 |
| Q15 | I speak slowly to foreign patients in simple Japanese. | -0.06 | -0.10 | **0.75** | 0.00 | 0.01 | 0.07 |
| Q16 | I use gestures to explain things to foreign patients who do not understand Japanese. | 0.12 | -0.11 | **0.69** | 0.04 | -0.09 | 0.19 |
| Q1 | I treat foreign patients with respect. | -0.07 | 0.30 | **0.58** | 0.11 | -0.14 | -0.15 |
| Q10 | I try to provide the same care to foreign patients as I do to Japanese patients. | -0.01 | 0.08 | **0.53** | -0.02 | 0.12 | -0.09 |
| Q23 | I observe the facial expressions and reactions of foreign patients to see if they understand what I want to convey. | 0.33 | -0.03 | **0.41** | 0.03 | 0.06 | 0.04 |
| Q37 | With the consent of the foreign patient, I share information with supporters. | 0.03 | -0.06 | 0.01 | **0.78** | -0.12 | 0.15 |
| Q36 | I cooperate with staff from various professions to meet the wishes of foreign patients. | -0.09 | -0.09 | 0.12 | **0.71** | 0.06 | 0.13 |
| Q33 | I share information with staff in my office in order to understand foreign patients. | 0.25 | 0.04 | -0.06 | **0.68** | -0.09 | -0.15 |
| Q35 | I do not handle foreign patients by myself, I collaborate with my colleagues in the office. | 0.02 | -0.09 | 0.21 | **0.65** | 0.00 | -0.06 |
| Q34 | I have detailed information on the daily lives of foreign patients in my medical records. | -0.08 | 0.20 | -0.15 | **0.53** | 0.10 | 0.10 |
| Q7 | When a foreign patient asks me a question about nursing care, I provide a detailed explanation. | 0.03 | -0.01 | -0.03 | -0.09 | **0.92** | -0.08 |
| Q6 | When I cannot satisfy the wishes of a foreign patient, I explain the reason and obtain his/her understanding. | -0.07 | -0.11 | -0.01 | 0.05 | **0.80** | 0.08 |
| Q8 | I listen to foreign patients carefully and try to understand their thoughts. | 0.01 | 0.22 | 0.28 | -0.03 | <u>**0.38**</u> | 0.08 |
| Q5 | I try not to show foreign patients that I am in hurry. | 0.02 | 0.22 | 0.00 | -0.05 | 0.29 | -0.05 |
| Q17 | I explain to foreign patients by showing them illustrations and drawings. | -0.05 | 0.13 | 0.02 | 0.08 | -0.01 | **0.65** |
| Q18 | I use a comparison chart for foreign patients in their native language or in an understandable foreign language and Japanese. | 0.00 | 0.12 | 0.12 | 0.11 | -0.01 | 0.36 |
| | Eigenvalues | 10.33 | 1.91 | 1.36 | 1.08 | 0.81 | 0.65 |
| | Percentage of variance | 32.29 | 5.98 | 4.25 | 3.37 | 2.53 | 2.05 |
| | Cumulative of total variance explained | 32.29 | 38.27 | 42.52 | 45.89 | 48.42 | 50.46 |

CCCSJN, Cross-cultural Competence Scale for Japanese Nurses.

https://nagoya.repo.nii.ac.jp/records/2008755

**Criterion-related validity of the CCCSJN.** The total CCCSJN score showed a significant positive correlation with the SCTO subscales "understanding diversity" and "understanding others" (understanding diversity: rs = 0.51, p < .001; understanding others: rs = 0.47, p < .001). The total score of CCCSJN also showed a significant positive correlation with the J-CCSS subscale "positive feelings toward different cultures" (rs = 0.69, p < .001; Table 6).

**Table 5. Factor loadings on the CCCSJN for Japanese nurses (N = 394).**

| 27 items (α = 0.94) | | 1 | 2 | 3 | 4 | 5 |
|---|---|---|---|---|---|---|
| **Factor 1: Cross-cultural understanding (α = 0.94)** | | | | | | |
| Q28 | I understand that foreign patients may have behavioral restrictions or rules according to their religion. | **0.84** | 0.18 | -0.58 | -0.11 | -0.03 |
| Q26 | I understand that different cultures have different values | **0.84** | 0.60 | 0.11 | -0.11 | -0.09 |
| Q27 | I understand that foreign patients' religion or beliefs may dictate traditional diets or dietary restrictions | **0.81** | 0.21 | -0.43 | -0.12 | -0.05 |
| Q31 | I have detailed information on the daily lives of foreign patients in my medical records. | **0.72** | -0.28 | 0.82 | 0.13 | 0.01 |
| Q29 | I understand that culturally sensitive nursing care must be provided to all foreign patients. | **0.72** | 0.15 | -0.69 | 0.01 | 0.04 |
| Q32 | I understand that foreign patients may have financial concerns. | **0.67** | -0.13 | -0.38 | 0.14 | 0.10 |
| Q25 | I understand that cultures of foreign patients may have different views regarding health and disease matters. | **0.64** | -0.15 | 0.29 | 0.03 | -0.07 |
| Q30 | I understand that foreign patients with a short period of stay may feel anxious about their hospital stay. | **0.62** | 0.39 | -0.11 | 0.12 | 0.14 |
| **Factor 2: Cross-cultural communication ability (α = 0.78)** | | | | | | |
| Q15 | I speak slowly to foreign patients in simple Japanese. | -0.04 | **0.71** | -0.08 | 0.05 | -0.03 |
| Q16 | I use gestures to explain things to foreign patients who do not understand Japanese. | 0.12 | **0.65** | -0.12 | 0.11 | -0.09 |
| Q1 | I treat foreign patients with respect. | -0.05 | **0.63** | 0.23 | 0.03 | -0.09 |
| Q10 | I try to provide the same care to foreign patients as I do to Japanese patients. | 0.02 | **0.54** | 0.02 | -0.04 | 0.13 |
| Q23 | I observe the facial expressions and reactions of foreign patients to see if they understand what I want to convey. | 0.33 | **0.40** | -0.07 | 0.07 | 0.04 |
| **Factor 3: Motivation for cross-cultural nursing (α = 0.82)** | | | | | | |
| Q13 | I want to be actively involved in the care of foreign patients. | -0.04 | 0.20 | **0.71** | -0.04 | -0.11 |
| Q9 | When I know I'll be dealing with foreign patients in advance, I try to research the culture of the country they come from. | 0.03 | -0.18 | **0.59** | 0.08 | -0.11 |
| Q2 | I frequently talk to foreign patients. | -0.12 | 0.36 | **0.51** | -0.02 | 0.08 |
| Q11 | I apply the experiences I have learned from foreign patients in my nursing care. | 0.17 | -0.02 | **0.48** | 0.01 | 0.02 |
| Q12 | When there is a difference between the wishes of foreign patients and the rules of the hospital, I think of other ways to address the issue together with the patient. | 0.09 | -0.13 | **0.45** | 0.08 | 0.18 |
| Q3 | I greet foreign patients in their native languages as much as possible. | -0.02 | 0.01 | **0.44** | -0.04 | 0.05 |
| **Factor 4: Cooperation with multiple professions (α = 0.88)** | | | | | | |
| Q37 | With the consent of the foreign patient, I share information with supporters. | 0.02 | 0.01 | -0.01 | **0.79** | -0.10 |
| Q36 | I cooperate with staff from various professions to meet the wishes of foreign patients. | -0.06 | 0.15 | -0.07 | **0.68** | 0.08 |
| Q33 | I share information closely with staff in my office in order to understand foreign patients. | 0.21 | -0.01 | 0.07 | **0.60** | -0.09 |
| Q35 | I do not handle foreign patients by myself alone, but collaborate with my colleagues in the office. | 0.01 | 0.25 | -0.03 | **0.58** | -0.01 |
| Q34 | I have detailed information on the daily lives of foreign patients in my medical records. | -0.07 | -0.10 | 0.21 | **0.55** | 0.11 |
| **Factor 5: Respect for foreign patients (α = 0.84)** | | | | | | |
| Q7 | When a foreign patient asks me a question about nursing care, I explain the reason in detail. | 0.04 | -0.04 | 0.01 | -0.11 | **0.92** |
| Q6 | When I cannot satisfy the wishes of a foreign patient, I explain the reason and obtain his/her understanding. | -0.03 | 0.00 | -0.07 | 0.11 | **0.70** |
| Q8 | I listen to foreign patients carefully and try to understand their thoughts. | 0.03 | 0.32 | 0.17 | -0.01 | **0.42** |
| | Eigenvalues | 8.78 | 1.77 | 1.44 | 1.00 | 0.78 |
| | Percentage of variance | 32.53 | 6.54 | 5.33 | 3.69 | 2.84 |
| | Cumulative of total variance explained | 32.53 | 39.07 | 44.40 | 48.09 | 50.92 |

CCCSJN, Cross-cultural Competence Scale for Japanese Nurses.

https://nagoya.repo.nii.ac.jp/records/2008756

**Internal consistency of CCCSJN.** The Cronbach's α for the overall CCCSJN was 0.94 (Table 5). Cronbach's α for each subscale of CCCSJN were 0.94 for "cross-cultural understanding," 0.78 for "cross-cultural communication ability," 0.82 for "motivation for cross-cultural nursing," 0.88 for "cooperation with multiple professions," and 0.84 for "respect for foreign patients." There was a strong correlation between the CCCSJN total score and all subscale factors (rs = 0.70–0.84, p < .001). Moderate correlations existed between each subscale (rs = 0.44–0.56, p < .001).

**Table 6. Correlations between total CCCSJN scores and SCTO and J-ISS subscales (n = 390).**

|  | CCCSJN | |
| --- | --- | --- |
|  | **r** | **p** |
| **SCTO: Understanding diversity (subscale)** | 0.51 | < .001 |
| **SCTO: Understanding other (subscale)** | 0.47 | < .001 |
| **J-ISS: Positive feelings toward different cultures (subscale)** | 0.69 | < .001 |

r, Spearman rank correlation coefficient; p, probability value; CCCSJN, Cross-cultural Competence Scale for Japanese Nurses; J-ISS, Japanese intercultural sensitivity scale; SCTO, Social critical thinking orientation.
https://nagoya.repo.nii.ac.jp/records/2008757

**CCCSJN test-retest reliability.** Of the 394 participants, 341 returned the questionnaire in the second study (response rate = 86.5%). Substantial correlations were found between the first and second studies (ICC [1,1] = 0.767 [0.72–0.81], p < .001).

## Discussion

This study developed a scale—the CCCSJN—to measure the cross-cultural competence of Japanese nurses to care for foreign patients. The scale demonstrated construct validity, criterion-related validity, internal consistency, and test-retest reliability.

The first factor is "cross-cultural understanding," which means understanding and accepting differences not only in cultures and customs but also in ways of thinking and values. Lack of understanding of cultural norms and traditions leads directly to inadequate information and poor-quality care [25]. Therefore, this scale indicates the importance of cross-cultural understanding to support patients' cultural beliefs and values for Japanese nurses who are not yet confident in foreign nursing. In previous studies, factors related to cross-cultural understanding were also a major construct [6, 18, 27, 28].

The second factor, "cross-cultural communication ability," comprises the communication skills of nurses so that foreign patients can receive healthcare without worry. Communication barriers reduce uptake and adherence to treatment and care and affect outcomes [26]. This factor included an easy-to-understand method to prevent misunderstandings due to language and cultural barriers and to check the degree of understanding. Communication ability was also a major component in previous studies [18, 27–29].

The third factor, "motivation for cross-cultural nursing," was the concept of nurses learning about cultural diversity and being positive about actively engaging with patients from different cultural backgrounds. In a previous study [27], a significant positive association was found between work attitude and motivation. Thus, it can be inferred that the concept of "motivation for cross-cultural nursing" promotes the ability of Japanese nurses to cope with cross-cultural situations. In Germany, a scale expressed as the concept of motivation and curiosity was developed [28], validated, and was deemed sufficiently reliable for adoption as a high-impact primary factor. This concept is also similar to the "cultural desires" of Campinha–Bacote's conceptual model. However, only a few scales adopted the concept of "cultural desires" [29, 30], and the validity and reliability were poorly tested.

The fourth factor, "cooperation with multiple professions," was not represented by existing scales in Europe, the United States, and Japan, confirming that this concept is unique to the CCCSJN. This means that nurses do not have to manage foreign patients on their own but can share information closely with staff at their workplace and collaborate with a wide range of professionals to ensure that they respond to the needs of foreign patients. Compared to the West, Japanese culture favors a mutually cooperative self-view [31, 32], valuing cooperative

relationships between individuals [33]. Japanese people are also more interested in group goals and cooperative behavior. Although "cooperation with multiple professions" is essential for nurses, it was particularly evident in the CCCSJN.

The fifth factor, "respect for foreign patients," refers to the attitude of responding to foreign patients with respect for their wishes and intentions. The Japanese are often characterized as having a national character that emphasizes respect for others, values compassion and courtesy, and may prioritize the needs of others over their own. In a 2019 Ministry of Land, Infrastructure, Transport, and Tourism attitude survey, more than 80% of respondents said it was important to be respectful and considerate of others [34]. "Respect for foreign patients" was an important attitude for nurses and a concept that emerged prominently in the CCCSJN.

"Utilization of social resources" was not included in the five-factor CCCSJN, although it was suggested that many Japanese nurses use social resources. "Use of a translation device," "cooperation with family and friends of patients who could understand Japanese," and "use of medical interpreters" were excluded at an early stage because of the ceiling effect. This meant that many Japanese nurses scored the highest.

Proof of criterion-related validity suggests that the CCCSJN could be an appropriate measure of cross-cultural competence. Total CCCSJN score was significantly positively correlated with the SCTO subscales "understanding diversity" and "understanding others." These questions were similar to those in the first factor of the CCCSJN, "cross-cultural understanding," and were useful for understanding foreign patients. Japanese nurses also tended to emphasize "understanding diversity" and "understanding others" even in their nursing practice for Japanese patients [22]. The total CCCSJN score also showed a significant positive correlation with the J-CCSS subscale "positive feelings toward different cultures," which was related to flexibility and adaptability in cross-cultural situations, and meaningful communication inspired by responses from others [23].

The CCCSJN had high internal consistency. Specifically, the Cronbach's α for all CCCSJN items was high at 0.94, and the five subscales also showed high internal consistency at 0.78–0.94, indicating high reliability. The other scales showed high internal consistency, similar to that of the CCCSJN. The J-CCCHS showed Cronbach's α for the five subscales ranged from 0.76–0.90 [6]. The Nursing Cultural Competence Scale had Cronbach's α for the four-subscale that ranged from 0.82–0.84 [35]. Moderate to strong correlations between the CCCSJN and subscales was also obtained (rs = 0.68–0.84, p < .001). Additionally, a strong positive correlation was found between the first and second studies on the test-retest reliability of the CCCSJN, indicating the stability of the scale.

## Study limitations

This study has five limitations. The first limitation is selection bias due to recruitment from a medical facility with a large number of foreign patients. In a multistage sampling study of 5430 nurses across Japan, 72.5% had experience caring for foreign patients [6] (compared with 82.5% in the current study). The number of foreign residents in Aichi Prefecture, where the current participants work, was second only to Tokyo; therefore, more people had experience in providing care for foreign patients than in other prefectures. Nurses working in areas with few foreign residents are expected to have less experience providing care for foreign patients. Since cross-cultural competence was significantly related to the experience of caring for foreign patients, the influence of locality on cross-cultural competence should also be carefully observed. It is difficult to generalize the present findings to all Japanese nurses. However, the results may be applicable to nurses in urban areas such as Kanto and Kansai.

Second, participants were about eight years younger and had eight years less nursing experience than nurses in Japan [36]. Educational attainment was 11.5% higher for nurses with diplomas and 7.3% lower for nurses with bachelor's degrees [36]. In previous studies, age and nursing experience were significantly related to cross-cultural competence, and educational background may be related to cross-cultural understanding learning and foreign language proficiency [37–39]. To confirm the accuracy of the CCCSJN, further studies with more-representative samples are needed.

Third, the structure of the CCCSJN did not include the use of social resources. In the item analysis, social resource use showed a ceiling effect in three of the four items and was deleted. It was considered that the utilization of social resources was not critical for nurses in Japan, and that it might permeate as usual nursing; however, future research could address this.

Fourth, recall bias could have occurred because some questions required recalling the content of care for foreign patients and the number of times they administered such care in a year. The recall period must be considered to avoid memory ambiguity and ensure accurate answers.

Fifth, female nurses were the only participants. The percentage of male nurses in Japan is 8.1%, which is about two times higher than 10 years ago [40], and this trend is expected to continue. Future studies including male nurses are warranted.

## Conclusions

This study was the first to use a semi-structured interview method to develop a scale of cross-cultural competence for Japanese nurses caring for foreign patients. The scale has a five-factor structure and demonstrated construct validity, criterion-related validity, internal consistency, and test-retest reliability. Therefore, the scale is useful for assessing cross-cultural competence. The CCCSJN can be employed to assess the cross-cultural competencies of Japanese nurses and identify those that need to be mastered, leading to improved cross-cultural competencies and care.

## Acknowledgments

We express our sincere appreciation to all the Japanese nurse from the six medical facilities in Aichi, Japan for their kind support and active cooperation in this study.

## Author Contributions

**Conceptualization:** Masako Sakamoto, Shigemi Iriyama, Yu Mon Saw.

**Data curation:** Masako Sakamoto, Shigemi Iriyama.

**Formal analysis:** Masako Sakamoto, Shigemi Iriyama, Yu Mon Saw.

**Funding acquisition:** Masako Sakamoto.

**Investigation:** Masako Sakamoto, Shigemi Iriyama, Yu Mon Saw.

**Methodology:** Masako Sakamoto, Shigemi Iriyama, Yu Mon Saw.

**Project administration:** Masako Sakamoto, Shigemi Iriyama.

**Resources:** Masako Sakamoto.

**Software:** Masako Sakamoto, Shigemi Iriyama.

**Supervision:** Masako Sakamoto, Shigemi Iriyama, Yu Mon Saw.

**Validation:** Masako Sakamoto, Shigemi Iriyama.

**Visualization:** Masako Sakamoto, Shigemi Iriyama.

**Writing – original draft:** Masako Sakamoto, Shigemi Iriyama, Yu Mon Saw.

**Writing – review & editing:** Masako Sakamoto, Shigemi Iriyama, Yu Mon Saw.

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
