## [Editor Report · Decision Letter 0]

29 Nov 2023

PONE-D-23-34062Developing and validating a cross-cultural competence scale for Japanese nursesPLOS ONE

Dear Dr. Sakamoto,

Thank you for submitting your manuscript to PLOS ONE. After careful consideration, we feel that it has merit but does not fully meet PLOS ONE’s publication criteria as it currently stands. Therefore, we invite you to submit a revised version of the manuscript that addresses the points raised during the review process.

We look forward to receiving your revised manuscript.

Kind regards,

Ali Khodi, PhD

Academic Editor

PLOS ONE

3. PLOS requires an ORCID iD for the corresponding author in Editorial Manager on papers submitted after December 6th, 2016. Please ensure that you have an ORCID iD and that it is validated in Editorial Manager. To do this, go to ‘Update my Information’ (in the upper left-hand corner of the main menu), and click on the Fetch/Validate link next to the ORCID field. This will take you to the ORCID site and allow you to create a new iD or authenticate a pre-existing iD in Editorial Manager. Please see the following video for instructions on linking an ORCID iD to your Editorial Manager account: https://www.youtube.com/watch?v=_xcclfuvtxQ".

5. Please include your tables as part of your main manuscript and remove the individual files. Please note that supplementary tables (should remain/ be uploaded) as separate ""supporting information"" files".

Additional Editor Comments:

Thank you for you submission. the results ection needs to be revised carefully. It does not present strong and enough suports for your study. Then the paper could be sent for reviewers.

---

## [Author Response · Author response to Decision Letter 0]

22 Jan 2024

Dr. Ali Khodi gave me useful advice. I thank him. In my case, I didn't understand the magazine's submission requirements, so his advice was helpful.

---

## [Decision Letter · Decision Letter 1]

5 Jun 2024

PONE-D-23-34062R1Developing and validating a cross-cultural competence scale for Japanese nursesPLOS ONE

Dear Dr. Sakamoto,

Thank you for submitting your manuscript to PLOS ONE. After careful consideration, we feel that it has merit but does not fully meet PLOS ONE’s publication criteria as it currently stands. Therefore, we invite you to submit a revised version of the manuscript that addresses the points raised during the review process.

We look forward to receiving your revised manuscript.

Kind regards,

Frantisek Sudzina

Academic Editor

PLOS ONE

Journal Requirements:

Reviewers' comments:

Reviewer's Responses to Questions

**Comments to the Author**

1. If the authors have adequately addressed your comments raised in a previous round of review and you feel that this manuscript is now acceptable for publication, you may indicate that here to bypass the “Comments to the Author” section, enter your conflict of interest statement in the “Confidential to Editor” section, and submit your "Accept" recommendation.

Reviewer #1: (No Response)

Reviewer #2: (No Response)

2. Is the manuscript technically sound, and do the data support the conclusions?

Reviewer #1: Partly

Reviewer #2: Yes

3. Has the statistical analysis been performed appropriately and rigorously? 

Reviewer #1: No

Reviewer #2: Yes

4. Have the authors made all data underlying the findings in their manuscript fully available?

Reviewer #1: Yes

Reviewer #2: Yes

5. Is the manuscript presented in an intelligible fashion and written in standard English?

Reviewer #1: (No Response)

Reviewer #2: Yes

6. Review Comments to the Author

Reviewer #1: This study developed a scale to measure cross-cultural competence among Japanese nurses. The scale demonstrated high internal consistency and test-retest reliability, as well as good cumulative explanatory power. However, it has several weaknesses that require revision.

The authors excluded outpatient and ward nurses. However, Table 1 includes nurses working in inpatient and outpatient departments. This creates a contradiction that needs clarification.

The authors did not have a clear definition of the concept of "cross-cultural competence" and how it differed from "cultural competence". They first mentioned that eight categories made up the concept. Later, they defined the concept based on a three-dimensional model proposed by Suh et al. The final scale consisted of five factors. There is a need to evaluate the construct validity of the scale against the original constructs.

The scales used to determine criterion-related validity were found to be weak. The SCTO (Social Critical Thinking Orientation) scale was developed specifically to measure critical thinking. J-ISS is a translated scale from a language other than Japanese and its factor structure differs from the original, indicating that it may not be a stable measurement. There may be more appropriate criteria for measuring cultural competence, as many scales already exist. Therefore, it is necessary to justify the use of current scales as criteria.

The authors used explanatory factor analysis indicating the concept of cross-cultural competence is ambiguous. Since there are already many cultural competence scales available, potential constructs have already been extensively discussed. Exploratory factor analysis is used when there is no hypothetical construct concept available. Therefore, it is recommended to speculate on the concept of cross-cultural competency and develop hypothetical constructs. When you use a construct concept, construct validity should be evaluated against the original constructs through confirmatory factor analysis. For this purpose, more rigorous methods, such as the maximum likelihood method or least squares method, are recommended.

The results of the explanatory factor analysis in this study were significantly different from the original possible constructs that included either 8 categories or 3 dimensions. This indicates that the construct validity has not been confirmed. Therefore, you need to discuss the reasons for this difference and provide a justification for it.

Additionally, please specify how CCSJN was measured, whether it was done using a 5-point Likert scale or a 7-point scale.

Elaborate discussions.

The author mentioned that item analysis was performed to confirm the construct validity of the CCCSJN. However, excluding items with a ceiling or floor effect does not necessarily confirm construct validity. It is important to evaluate the factor structures against the original construct concept. The original eight categories included dimensions such as using social resources, ingenuity in explanation, and financial support. However, these items were excluded because of the ceiling effect. The question then arises: are current Japanese nurses competent in introducing social resources or advice in financial support that is appropriate for foreign patients? Additionally, is it valid to exclude these constructs from cross-cultural competence, or is it just a problem with the questionnaire wording?

The authors noted that the existing scales in Europe and the United States did not include the aspect of "collaboration with multiple professions". However, this aspect may not be specific to nurses taking care of foreign patients. It could be because this construct is too general and not included in existing scales. Since the nurses mentioned that this aspect was important in the qualitative study, it raises the question of why it was excluded from the competency.

Reviewer #2: Please review 12 minor comments to the attached manuscript from pages 9-33.

Page 9 - are these actual numbers of percentage?

Page 12 - what is the ultimate purpose of developing and validating the Cross-Cultural Competence Scale for Japanese Nurses? This will highlight the usefulness of this study which can be added to the recommendations.

add: "in caring for patients with different cultural backgrounds."

Page 13 - what were these three items that were removed?

Page 16 - How were these variables treated? You may include observations such as the group being homogenous (or not) and how this likely affected the results and implications on the generalizability of the findings.

You may also add here how cross-cultural variables were used in the study, esp in the analysis of findings.

Page 18 - This is the main study, right? If so, just edit to say, "In the main study" (this is to differentiate from the preliminary part where the initial items for the questionnaire were developed.

Page 27 - of variance on cultural competence.

Page 29 -

please compute again... this seems to be 86.5% response rate

replace with "competence"

Why are the first two factors not discussed here? Cross-cultural understanding and communication ability are important factors and explain the highest variance in cultural competence of Japanese nurses. I suggest that these should also be mentioned in the discussion and how these are compared with other studies.

Page 33

You may also look into the socio-demographic characteristics and cross-cultural experiences of the sample in the study for sources of study limitations.

7. PLOS authors have the option to publish the peer review history of their article (what does this mean?). If published, this will include your full peer review and any attached files.

Reviewer #1: No

Reviewer #2: No

---

## [Author Response · Author response to Decision Letter 1]

2 Sep 2024

Response to Reviewer 1

Manuscript ID: PONE-D-23-34062 

We sincerely appreciate your comments on our manuscript. We revised and improved our manuscript accordingly. Please see our point-to-point responses below. Our responses here and our revisions in the manuscript are represented in blue. 　　　　　　　　　　　　　　　　　　　　　　　　　　　　　　　　　　　　　　　　　 

Question 1 The authors excluded outpatient and ward nurses. However, Table 1 includes nurses working in inpatient and outpatient departments. This creates a contradiction that needs clarification.　　　 　　　　　　　　　　　　　　　　　　　　　　　　　　　　　　　　　　　　　　　　　　　　　　　　　　　　　　　　　 

Response: We revised for additional clarity (lines 126–127): “Head nurses of outpatient departments and wards were excluded because administrative responsibilities greatly expanded their roles beyond simply providing care.”

Question 2

The authors did not have a clear definition of the concept of "" and how it differed from "cultural competence". They first mentioned that eight categories made up the concept. Later, they defined the concept based on a three-dimensional model proposed by Suh et al. The final scale consisted of five factors. There is a need to evaluate the construct validity of the scale against the original constructs.

Response: We operationally defined “cross-cultural competence” by referencing Suh’s three-dimensional model. The concept of the scale was extracted by an exploratory factor analysis based on the primary study. We revised as follows (lines 143–146): “‘Cross-cultural competence’ is the ability to provide appropriate care to foreign patients with different cultural backgrounds. It was defined with reference to the three-dimensional model of Suh et al. [13]. This ability requires cross-cultural understanding, communication skills, and respect for foreign patients.”

Question 3

The scales used to determine criterion-related validity were found to be weak. The SCTO (Social Critical Thinking Orientation) scale was developed specifically to measure critical thinking. J-ISS is a translated scale from a language other than Japanese and its factor structure differs from the original, indicating that it may not be a stable measurement. There may be more appropriate criteria for measuring cultural competence, as many scales already exist. Therefore, it is necessary to justify the use of current scales as criteria. 

Response: Of the seven STCO subscales, “understanding others” and “understanding diversity” were similar to those of the CCCSJN. For nurses, the ability to understand people is important for practical nursing skills (Matsutani, 2010). The STCO is a measure of critical thinking, but it comprises not only skills but also attitude. Attitude refers to “having an open mind and flexibility to listen and understand others and seek out information, knowledge, and choices” (Kusumi 2015), which is important in nursing practice and in the examination of cross-cultural correspondence ability. For these reasons, the STCO was adopted as an assessment tool for criterion-related validity. The results of the analysis also showed that the CCCSJN was positively correlated with understanding others and diversity (rs = 0.47, p < .001 and rs = 0.51, p < .001, respectively).　　　　　　　　　　　　　　 

Although the J-ICC differed from the five-factor structure of the ISS, a confirmatory factor analysis confirmed the suitability of the three-factor structure. The first factor, positive intercultural feelings, was adopted in this study because it is important for intercultural competence. Results also showed that the CCCSJN was significantly correlated with positive feelings toward other cultures (rs = 0.69, p < .001).

In conclusion, the STCO and J-ICC are appropriate scales for evaluating the criterion-related validity of the CCCSJN.

References

Matsutani M, Miura Y, Hirabayashi Y, et al. Nursing competency: Concept, structure of dimensions, and assessment. J St Lukes Soc Nurs Res. 2010;14: 18-28. http://arch.slcn.ac.jp/dspace/bitstream/10285/6257/2/2010031-gakkai14 (2)-6257.pdf

Takashi K. Critical thinking in education: A study based on the nursing process. J Jpn Soc Nurs Diag. 2015;20: 33-38. http://hdl.handle.net/2433/198120　　

Suzuki Y. Development of intercultural sensitivity scale in Japan. Bull Grad School Human Devel Environ Kobe Univ. 2016;9: 39-44. https://cir.nii.ac.jp/crid/1390572174883505792.

Fritz W, Graf A, Hentze J, Möllenberg A, Chen GM. An examination of Chen and Starosta’s model of intercultural sensitivity in Germany and United States. Intercult Commun Stud. 2005;16: 53-65. https://www.doi.org/10.5815/ijmecs.2015.06.01

Kuwano N, Fukuda H, Murashima S. Factors affecting professional autonomy of Japanese nurses caring for culturally and linguistically diverse patients in a hospital setting in Japan. J Transcult Nurs. 2016;27: 567-73. https://www.doi.org/10.1177/1043659615587588

Toda T, Maru M. Cultural sensitivity of Japanese nurses: Exploring clinical application of the Intercultural Sensitivity Scale. Open J Nurs. 2018;8: 640-55. https://doi.org/10.11477/mf.700420004

Question 4

The authors used explanatory factor analysis indicating the concept of cross-cultural competence is ambiguous. Since there are already many cultural competence scales available, potential constructs have already been extensively discussed. Exploratory factor analysis is used when there is no hypothetical construct concept available. Therefore, it is recommended to speculate on the concept of cross-cultural competency and develop hypothetical constructs. When you use a construct concept, construct validity should be evaluated against the original constructs through confirmatory factor analysis. For this purpose, more rigorous methods, such as the maximum likelihood method or least squares method, are recommended.

Response: The CCCSJN is based on 37 questionnaire items identified in the primary study. An exploratory factor analysis was performed, and the principal factor method was chosen as the extraction method to thoroughly explore the underlying structure. This approach allowed us to effectively capture the unique aspects of cross-cultural competence in the Japanese context.

Question 5 

The results of the explanatory factor analysis in this study were significantly different from the original possible constructs that included either 8 categories or 3 dimensions. This indicates that the construct validity has not been confirmed. Therefore, you need to discuss the reasons for this difference and provide a justification for it.

Response: The CCCSJN is based on an analysis of 37 questionnaire items identified in the primary study. The five factors extracted through an exploratory factor analysis reflect the characteristics of 393 Japanese nurses. The CCCSJN scale was not developed based on the original three-dimensional model, as it was essential to account for the unique cultural and professional context of Japanese nurses. The observed differences in factor structure highlight the distinct aspects of cross-cultural competence in this setting. This tailored approach was necessary to accurately capture these specific competencies, thereby justifying the divergence from the original constructs. Future research will aim to further validate these findings through a confirmatory factor analysis.

Question 6

Additionally, please specify how CCSJN was measured, whether it was done using a 5-point Likert scale or a 7-point scale.

Response: We added the following sentence (lines 113–114): “The CCSJN used a five-point Likert scale ranging from “not at all” to “very true.”

Question 7 

The author mentioned that item analysis was performed to confirm the construct validity of the CCCSJN. However, excluding items with a ceiling or floor effect does not necessarily confirm construct validity. It is important to evaluate the factor structures against the original construct concept. The original eight categories included dimensions such as using social resources, ingenuity in explanation, and financial support. However, these items were excluded because of the ceiling effect. The questions then arise: are current Japanese nurses competent in introducing social resources or advice in financial support that is appropriate for foreign patients? Additionally, is it valid to exclude these constructs from cross-cultural competence, or is it just a problem with the questionnaire wording? 

Response: The CCCSJN is a measure analyzed by an exploratory factor analysis. The concepts of the nine categories mentioned above are only tentative concepts named from 37 items extracted from　the primary study. As you pointed out, we performed the item analysis first and excluded the inappropriate items. Of the four items related to the use of social resources, three were excluded owing to ceiling effects. However, we needed to see if it was appropriate to exclude them from the CCCSJN in the first place. Therefore, for confirmation, an exploratory factor analysis was performed on 37 items in the CCCSJN draft before item analysis. We clarified as follows (lines 267–272): “The results were obtained by an exploratory factor analysis following the analysis of 37 items in the draft CCCSJN. To confirm the factor structure, an exploratory factor analysis was performed on the draft before item analysis. The following 10 items with factor loadings < 0.4 were excluded: Q5, Q14, and Q17–Q24. The four items related to the utilization of social resources were included. A five-factor structure comprising 27 items was clarified.”

Question 8

The authors noted that the existing scales in Europe and the United States did not include the aspect of "collaboration with multiple professions". However, this aspect may not be specific to nurses taking care of foreign patients. It could be because this construct is too general and not included in existing scales. Since the nurses mentioned that this aspect was important in the qualitative study, it raises the question of why it was excluded from the competency. 

Response: As suggested, cooperation with multiple professions is an essential skill for nurses and is possibly not limited to cross-cultural competence. Therefore, we think it was not included in the Western scale. However, it was included in the CCCSJN because it was reflects Japanese characteristics. Compared to Westerners, the Japanese are expected to value cooperation, maintain harmony with their peers, and take appropriate actions. Further, Japanese nurses are not used to caring for foreign patients. For a Japanese nurse to provide appropriate nursing care to a foreign patient, it is important to work cooperatively with multiple professions.

In addition to the above, the following revisions were made:

Added “care” on line 227. 

Changed “three” to “five” on line 367.

Added the following to lines 386–390: “Third, the structure of the CCCSJN did not include the use of social resources. In the item analysis, social resource use showed a ceiling effect in three of the four items and was deleted. It was considered that the utilization of social resources was not critical for nurses in Japan, and that it might permeate as usual nursing; however, future research could address this.”　　　　　　　　　　　　　　　　　　　　　　　　　　

Changed “second” to “fourth” on line 391 and “third” to “fifth” on line 395.　

We carefully reviewed your comments and made key revisions when necessary. Please let us know if there are any further concerns. We look forward to hearing from you.

Kind regards,

Masako Sakamoto (PONE-D-23-34062)

Response to Reviewer 2

Manuscript ID: PONE-D-23-34062 

We sincerely appreciate your comments on our manuscript. We revised and improved our manuscript accordingly. Please see our point-to-point responses below. Our responses here and our revisions in the manuscript are represented in red.

Question 1

Page 9 - are these actual numbers of percentage?

Response: We revised as follows (line 46): “from 24.3% to 50.2%.”

Question 2

Page 12 - what is the ultimate purpose of developing and validating the Cross-Cultural Competence Scale for Japanese Nurses? This will highlight the usefulness of this study which can be added to the recommendations

Response: We clarified as follows (lines 93–100): “The ultimate purpose of developing and validating the Cross-cultural Competence Scale for Japanese Nurses (CCCSJN) was to create a tool that accurately visualizes the current state of cross-cultural competency among Japanese nurses. This will provide valuable data to guide improvements in cross-cultural competency, enhancing the ability of healthcare practitioners to effectively serve Japan’s growing and increasingly diverse foreign population, which has been expanding. As Japan continues to welcome more foreigners, cultural competence in healthcare is becoming increasingly essential, and this study helps to close that gap.

Question 3

add: "in caring for patients with different cultural backgrounds."

Response: We revised accordingly (lines 104–105): “To develop the CCCSJN, it was necessary to use questionnaire items that reflect the characteristics of Japanese nurses in caring for patients with different cultural backgrounds.”

Question 4

Page 13 - what were these three items that were removed?

Response:　The three excluded items were related to the utilization of social resources, confirmation, and dignity. We clarified as follows (lines 116–119): “The three items related to utilization of social resources, confirmation, and dignity were deleted because they were redundant with other questions or were difficult to understand, leaving 37 questions in the final draft.”

Question 5

Page 16 - How were these variables treated? You may include observations such as the group being homogenous (or not) and how this likely affected the results and implications on the generalizability of the findings.

Response: We treated the variables using the following steps, and descriptive statistics were used for variables related to the social attributes of the participants to understand their background. The median age of the participants was 33 years, and the median nursing experience was 9.8 years. The most common age of nurses in Japan was 40~49 years (mean = 41.3 years). The number of years of nursing experience ranged from 10 to 19 years (mean = 17.8 years). Regarding educational background, 83.7% had diplomas, and 11.2% held bachelor’s degrees. However, the educational background of nurses in Japan as a whole was 72.2% and 18.5%, respectively.

We revised as follows for additional clarity (lines 379–385): “Second, participants were about eight years younger and had eight years less nursing experience than nurses in Japan. Educational attainment was 11.5% higher for nurses with diplomas and 7.3% lower for nurses with bachelor’s degrees. In previous studies, age and nursing experience were significantly related to cross-cultural competence, and educational background may be related to cross-cultural understanding learning and foreign language proficiency. To confirm the accuracy of the CCCSJN, further studies with more-representative samples are needed.”

We also added four new references:

36. Japan Nursing Association. 2021 Fact-finding Survey of Nursing Staff: Japan Nursing Association Survey Research Report No. 98; 2022. https://www.98.pdf (nurse.or.jp)

37. Li J, He Z, Luo Y, Zhang R. Perceived transcultural self-efficacy of nurses in general hospitals in Guangzhou, China. Nurs Res. 2016;65: 371-379. https://www.doi.org/10.1097/NNR.0000000000000174

38. Tsui-Ting L, Miao-Yen C, Yu-Mei C, Mei-Hsiang L. A preliminary study on the cultural competence of nurse practitioners and its affecting factors. http://www.dx.doi.org/10.3390/healthcare10040678

39. Hietapakka L, Elovainio M, Wesolowska K, Aalto AM, Kaihlanen AM, Sinervo T, Heponiemi T. Testing the psychometric properties of the Finnish version of the cross-cultural competence instrument of healthcare professionals (CCCHP). BMC Health Serv Res. 2019;19: 294. https://www. doi.org.10.1186/s12913-019-4105-2

Question 6

You may also add here how cross-cultural variables were used in the study, esp in the analysis of findings.

Response: A

---

## [Editor Report · Decision Letter 2]

5 Sep 2024

Developing and validating a cross-cultural competence scale for Japanese nurses

PONE-D-23-34062R2

Dear Dr. Sakamoto,

We’re pleased to inform you that your manuscript has been judged scientifically suitable for publication and will be formally accepted for publication once it meets all outstanding technical requirements.

Kind regards,

Frantisek Sudzina

Academic Editor

PLOS ONE
---

## [Editor Report · Acceptance letter]

18 Oct 2024

PONE-D-23-34062R2 

PLOS ONE

Dear Dr. Sakamoto, 

I'm pleased to inform you that your manuscript has been deemed suitable for publication in PLOS ONE. Congratulations! Your manuscript is now being handed over to our production team.

Kind regards, 

on behalf of

Dr. Frantisek Sudzina 

Academic Editor

PLOS ONE